# Bisimidazolium Salt Glycosyltransferase Inhibitors Suppress Hepatocellular Carcinoma Progression In Vitro and In Vivo

**DOI:** 10.3390/ph15060716

**Published:** 2022-06-05

**Authors:** Xue Luan, Ming Sun, Xue Zhao, Jingyi Wang, Ye Han, Yin Gao

**Affiliations:** Key Laboratory for Molecular Enzymology and Engineering of Ministry of Education, School of Life Sciences, Jilin University, Changchun 130012, China; luanxue19@mails.jlu.edu.cn (X.L.); sunming19@mails.jlu.edu.cn (M.S.); zhaoxue21@mails.jlu.edu.cn (X.Z.); jingyiw20@163.com (J.W.); hanye21@mails.jlu.edu.cn (Y.H.)

**Keywords:** glycosyltransferase inhibitor, anticancer agents, cytotoxicity, ER stress, cell-cycle and apoptosis

## Abstract

Hepatocellular carcinoma is a leading cause of cancer death, and the disease progression has been related to glycophenotype modifications. Previously synthesized bisimidazolium salts (C20 and C22) have been shown to selectively inhibit the activity of glycosyltransferases in cultured cancer cell homogenates. The current study investigated the anticancer effects of C20/C22 and the possible pathways through which these effects are achieved. The therapeutic value of C20/C22 in terms of inhibiting cancer cell proliferation, metastasis, and angiogenesis, as well as inducing apoptosis, were examined with hepatic cancer cell line HepG2 and a xenograft mouse model. C20/C22 treatment downregulated the synthesis of SLe^x^ and Le^y^ sugar epitopes and suppressed selectin-mediated cancer cell metastasis. C20/C22 inhibited HepG2 proliferation, induced cell-cycle arrest, increased intracellular ROS level, led to ER stress, and eventually induced apoptosis through the intrinsic pathway. Furthermore, C20/C22 upregulated the expressions of death receptors DR4 and DR5, substantially increasing the sensitivity of HepG2 to TRAIL-triggered apoptosis. In vivo, C20/C22 effectively inhibited tumor growth and angiogenesis in the xenograft mouse model without adverse effects on major organs. In summary, C20 and C22 are new promising anti-hepatic cancer agents with multiple mechanisms in controlling cancer cell growth, metastasis, and apoptosis, and they merit further development into anticancer drugs.

## 1. Introduction

Hepatocellular carcinoma is closely related to advanced liver diseases and is one of the leading life-threatening cancers worldwide [1]. Despite the advances in medical treatments, the emergence of drug resistance [2,3] and tumor recurrence have impeded efforts to reduce cancer mortality [4,5]. Therefore, the discovery of targeted low-toxicity drugs for the treatment of hepatocellular carcinoma (HCC) is still urgently needed.

Glycosylation has an essential role in physiological and pathological processes [6,7]. Cancer progression associated with abnormal glycosylation and these altered glycans regulate a series of pathological processes, including cell–cell interaction, cell adhesion, receptor activation, and signal transduction [8]. Due to the key role of glycosylation in protein expression and transport, ligand binding, and receptor signal transmission [9], the study of its involvement in signal pathway disorders and the development of new strategies to modify cell glycosylation have received significant attention. Recent research has indicated that glycosylation modifications are closely related to HCC, and the upregulation of galactosyltransferase expression was shown to be related to cancer metastasis [10]. Glycosyltransferases (GTs) catalyze the transfer of glycosyl groups from activated donor molecules to specific acceptor molecules and influence the biosynthesis of sugar chains [11]. Tumor cells overexpress specific glycosyltransferases which promote the formation of tumor glycans and facilitate cancer progression [12,13]. We established a connection between the B4GALT family and microtubule spindle assembly in HCC and reported that B4GALT4 is a critical promotor for HCC [14]. Therefore, glycosyltransferase inhibitors that selectively inhibit the glycosyltransferases in cancers are promising candidates for novel anticancer agents [15].

The galactosyltransferase (GalT) family encodes a set of type II transmembrane glycoproteins, which can transfer galactose to the *N*-acetyl-glucosamine acceptor to produce a lactosamine (LacNAc) chain [16] (Figure 1). LacNAc is present on glycoproteins and glycolipids to serve as a scaffold for the synthesis of terminal sugar epitopes such as blood group antigens and Lewis structures [17,18,19]. ꞵ4GalT (Figure 1) has been identified as an apoptosis-related gene in some cancer cells [19], associated with anticancer drug resistance [20,21] and the regulation of microtubule spindle assembly [14]. Other members of this family (such as ꞵ3GalT (Figure 1)) are also involved in the activation of signal transduction pathways, the prevention of apoptosis, and the promotion of metastasis in breast cancer [22]. Polypeptide-GalNAc transferase (ppGalNAcT) is the initiating enzyme for mucin-type *O*-glycosylation (Figure 1) [23]. Overexpression of ppGalNAcT is closely associated with the occurrence, metastasis, and poor prognosis of breast cancer [24]. Evidence has revealed that core 2 ꞵ1,6-*N*-acetylglucosaminyl transferase 1 (C2GnT1) synthesizes the core 2 branching structure on *O*-glycans (Figure 1) and is associated with biological aggressiveness of endometrial carcinoma cells [25]. The overexpression of the terminal monosaccharide, such as sialic acid, is closely related to the malignancy and metastasis of cancer [26]. In contrast with normal cells, cancer cells often overexpress the sialyl-Lewis x (SLe^x^) sugar epitope on glycoproteins or glycosphingolipids (Figure 1) on the surfaces of cells [27]. In hepatic cancer cells, SLe^x^ is highly expressed on core 2 branched *O*-glycans, where SLe^x^ can be recognized by selectins on the surface of endothelial cells [28]. The interaction of selectins with the glyco-ligands mediates the transport of cancer cells to the blood or lymphatic circulatory system, and these cells then reach the new host organ through the blood or lymphatic system to complete the process of cancer cell metastasis [29,30]. Therefore, inhibiting the synthesis of glyco-ligands for selectins has become a promising approach to block cancer metastasis [31].

One of the glycosyltransferase inhibitors that has been widely studied is the *N*-glycosylation inhibitor tunicamycin, which has the ability to inhibit cancer cell growth and reduce aggressiveness [32,33]. Tunicamycin was shown to inhibit the growth of breast and liver cancer cells by inhibiting metastasis and inducing apoptosis [34,35]. Studies have shown that *O*-glycosylation inhibitors can also induce apoptosis [36,37] and block cancer cell invasion [38]. However, these inhibitors inhibit the initial step of *N*/*O*-glycosylation and are also toxic to normal cells [39,40]. The previously synthesized bisimidazolium salts C20 and C22 (Figure 2A) were reported to selectively inhibit the activities of GalT, ppGalNAcT, and C2GNT1 in cultured cancer cells [41]. These glycosyltransferases have been found highly expressed in hepatic cancer [7,28,42]. On this basis, we further evaluated the anti-hepatic cancer effects and determined the possible pathways through which C20/C22 provides anticancer effects using HepG2 as a cell-based model.

## 2. Results

### 2.1. C20/C22 Inhibited HepG2 Cell Proliferation, Adhesion, Migration, and Invasion

The synthesized C20 and C22 bisimidazolium salts (Figure 2A) are non-substrate analog glycosyltransferase inhibitors [41] that can potentially inhibit the targeted glycosyltransferases in cell cultures to modify cancer cell glycosylation, which is closely related to metastasis [38,43,44,45,46,47]. The HepG2 cell line that highly expresses the targeted glycosyltransferases (galactosyltransferase (GalT), polypeptide-*N*-acetyl-galactosaminyltransferase (ppGalNAcT), and core 2 synthase (C2GNT1) [28,48,49,50]) was selected as the cell model to investigate the anticancer activities of C20 and C22.

The cell viability of HepG2 gradually decreased with increasing concentrations of C20 and C22, and the half-maximal growth-inhibitory concentrations (IC_50_) of C20 and C22 were 2.09–2.38 μM (95% CI) and 1.92–2.31 μM (95% CI), respectively (Figure 2B,C). HepG2 cells were more sensitive to C20 or C22 treatment in comparison to SMMC-7721 cells; thus, HepG2 was selected as the cell model for the subsequent research (Figure 2B and Appendix A). In contrast, the cell viability of HUVEC and HEK293A cells were not significantly affected by the C20/C22 treatment at low concentrations, thus suggesting that C20 and C22 can selectively inhibit the proliferation of HepG2 cells.

Pretreating HepG2 with 2 μM C20/C22 for 24 h effectively inhibited the adhesion of HepG2 cells to HUVEC cells in comparison with the untreated control group (Figure 2D). Trans-well assays with Matrigel loaded onto the inserts were used to simulate the endothelial system in vitro. HepG2 cells that were treated with C20/C22 for 24 h showed a significant reduction in susceptibility to trans-endothelial invasion (Figure 2E). HepG2 cell viability was hardly affected by the treatment of C20/C22 at 1 μM (Figure 2B), but the cell migration rate was significantly inhibited (Figure 2F–I). As the administration concentration was increased, the migration inhibitory effect became more pronounced.

### 2.2. C20/C22 Modified Cell Surface Glycosylation and Inhibited Binding to Selectins

Lectins are a group of carbohydrate recognition proteins, which serve as important tools for glycan structure analysis [51,52]. Because of the specificity in binding of unique glycan linkage and monosaccharide units, lectins were used in the current study to identify the unique glycan structural properties of HepG2 after C20/C22 treatment. Herein, lectins Ricinus communis I (RCA I), *Maackia amurensis* II (MAL II), *Sambucus nigra* (SNA), Concanavalin A (Con A), Phaseolus vulgaris Leucoagglutinin (PHA-L), wheat germ (WGA), and *Maackia amurensis* I (MAL I) were used to recognize the glycan structures Gal, Neu5Ac(α2-3), Neu5Ac(α2-6), *N*-glycan mannose core, Galβ4GlcNAcβ6, GlcNAc, and Galβ4GlcNAc, respectively. C20/C22 treatment reduced the expression of Gal residue, as well as Galβ4GlcNAcβ6 and Galβ4GlcNAc units on the sugar chain, while increasing the presence of GlcNAc (Figure 3). The level of Neu5Ac(α2-3) was decreased in the C22-treated groups, but Neu5Ac(α2-6) remained unchanged in both C20 and C22 treatment groups (Figure 3).

C20/C22 was also found to affect the expression of Lewis structures (Figure 4A–D). The expressions of SLe^x^ and Lewis y (Le^y^) on the surfaces of HepG2 cells were dramatically reduced after C20/C22 treatment compared with the untreated control groups. In contrast, the expressions of Lewis a (Le^a^) and Lewis b (Le^b^) were increased after treatment with C20/C22.

SLe^x^ is a ligand that binds with selectins [53] and is involved in cancer cell metastasis [47,54,55]. Treating HepG2 with C20/C22 significantly reduced the number of cells that bind to E- and P-selectin in a dose-dependent manner (Figure 4E–H). Consistently, HepG2 treated with neuraminidase (which enzymatically cleaved the sialic acid residues from the surfaces of cells) reduced the binding of cells to E- and P-selectin (Figure 4I,J).

### 2.3. C20/C22 Increased the Susceptibility of HepG2 to TRAIL-Induced Apoptosis

Death receptors such as Fas, DR4, and DR5 belong to the tumor necrosis factor (TNF) superfamily and are important regulators of cell survival, apoptosis, immune response, and tumor growth [56]. As indicated by the Western blot (WB) analysis (Figure 5A), the expressions of DR4 and DR5 increased after incubation with C20/C22. The gray value calculation for the WB analysis showed an upregulation of DR4 and DR5 expressions along with the administration time in comparison to the 0 h groups with statistical significance (Figure 5B). The immunofluorescence staining reflected the intracellular and cell surface distribution of DR4 and DR5. For the cells treated with C20/C22 for 24 h, the expression of DR4 and DR5 increased significantly both intracellularly and on the cell surface (Figure 5C,D). Binding of TRAIL to DR4 and DR5 receptors induces cell apoptosis [57].

Cells pretreated with C20/C22 for 24 h were continuously incubated for 24 h in the presence of 30 ng of TRAIL per well, and the apoptotic cells were detected by flow cytometry. Without the pretreatment with C20/C22, no significant difference was observed between the TRAIL-treated group and the control group (Figure 5E,F). In contrast, TRAIL-induced apoptosis was increased by 40% among the cells pre-treated with 2 μM C20/C22 in comparison with the TRAIL-treated group. C20/C22 treatment also led to an increased expression of Fas (Figure 5G,H). The immunofluorescence results revealed that the Fas receptor only increased intracellularly, but its cell surface distribution was not significantly affected compared with the untreated cells (Figure 5I). Cells treated with C20/C22 showed no significant alterations in apoptosis after the addition of anti-Fas antibody (CH11) in comparison to the cells treated with CH11 alone (Figure 5J,K).

### 2.4. Treatment with C20/C22 Induced Endoplasmic Reticulum (ER) Stress and Cell-Cycle Arrest

Endoplasmic reticulum stress (ER stress) is mainly induced by the signal pathways through inositol requiring enzyme 1 (IRE1), protein kinase RNA-activated-like ER kinase (PERK), and activating transcription factor 6 (ATF6). The expression of the ER stress marker protein glucose-regulated protein 78 (GRP78) was upregulated in response to the treatment with C20/C22, while the expressions of ER membrane proteins IRE1, ATF6, and PERK were also increased (Figure 6A,B). In addition, phosphorylation of the IRE1 downstream protein JNK was upregulated. Normally, ER stress led to cell-cycle arrest at G2/M; thus, the HepG2 cell growth cycle after C20/C22 treatment was analyzed via flow cytometry. Compared with the control group, treatment with 3 μM of C20/C22 increased the ratio of cells at the G2/M phase (Figure 6C,D).

### 2.5. C20/C22 Induced Mitochondrial Dysfunction and Intrinsic Apoptosis of HepG2 Cells through the ER Stress Pathway

Treatment with C20/C22 induced ER stress of HepG2 cells, leading to the increased transcription of CHOP (Figure 6A) in comparison with the untreated groups and this increased expression was time-dependent. CHOP also acts as a transcription regulator to upregulate the expressions of Bax and Bak (Figure 7A,B), which can pierce the mitochondrial outer membrane to reduce the mitochondrial membrane potential (MMP) (Figure 7C). This eventually led to the release of cytochrome c (Cyto C) (Figure 7A,B) and ROS. DCFH-DA was used to determine the level of intracellular ROS, which was significantly increased due to C20/C22 treatments (Figure 7D).

Cells under ER stress and mitochondrial dysfunction could undergo apoptosis, leading to the orderly and effective removal of damaged cells [58]. To further explore the effects of C20/C22 on apoptosis-associated proteins, WB analysis was performed to determine the expressions of anti- and proapoptotic proteins. Treatment with C20/C22 altered the ratio of proapoptotic proteins (caspase 8, 9) and antiapoptotic proteins (Bcl-2) (Figure 8A–C). In addition, treatment with C20/C22 increased the expression of apoptosis effector caspase 3/7 (Figure 8D), leading to the activation of caspase 3 (cleaved caspase 3) (Figure 8A–C) and causing cells to undergo apoptosis through the intrinsic pathway (Figure 8E,F).

### 2.6. C20 and C22 Exhibit Antitumor Effects In Vivo without Causing the Histopathological Changes in Major Organs

The in vivo toxicity study of C20/C22 in healthy nude mice showed no significant body weight reduction when the injection dosage was 5 μmol/kg or lower (Figure 9A). Tumor-bearing mice that were treated with C20/C22 (5 μmol/kg) showed reduced tumor volume (Figure 9B,C) and tumor weight (Figure 9D) (*p* < 0.001), while the body weight was barely affected (Figure 9E). There was no significant difference between the groups in the ratio of the mass of major organs to the body weight of mice (Figure 9F). The blinded assessment of H&E staining revealed that no significant histopathological changes in major organs of mice injected with 5 μmol/kg C20/C22 in comparison to the control group mice that were injected with PBS (Figure 9G). The immunohistochemistry stain of the major organs with antibodies of GRP78, Cyto C, and cleaved caspase 3 showed no significant difference between the C20/C22-treated groups and the control groups, further proving the low toxicity of C20/C22 (Appendix A).

The immunohistochemical analysis of tumor tissues indicated that C20/C22 treatment led to ER stress in tumor, which could be attributed to the increased expression of GRP78 (Figure 10A). The increased level of Cyto C (Figure 10B) in the treatment groups suggested that the administration of C20/C22 led to mitochondrial dysfunction in tumor cells. Elevated expression of cleaved caspase 3 (Figure 10C) suggested that C20/C22 induced apoptosis in tumor cells. The expressions of death receptors DR4 (Figure 10D) and DR5 (Figure 10E) were increased upon C20/C22 treatment. Furthermore, C20/C22 treatment decreased the expression of SLe^x^ in tumor tissues, thus reducing tumor metastasis through decreasing the interactions of tumor cells with selectins (Figure 10F). Taken together, results from the in vivo experiments are consistent with the results obtained in the in vitro studies. In addition, C20/C22 treatment inhibited tumor angiogenesis (Figure 10G) and tumor cell proliferation (Figure 10H), which could be attributed to the decreased expression of CD31 and Ki67, respectively.

## 3. Discussion

Hepatocellular carcinoma is one of the most common life-threatening diseases with complex cellular glycosylation modifications during the progression [6]. Aberrant glycosylation in cancers can be used as potential cancer biomarkers, and these cancer-associated glycans and their synthesizing enzymes have shown great potential as targets in the development of anti-cancer therapy [59].

This study unraveled the effect of bisimidazolium salt glycosyltransferase inhibitors C20 and C22 as anticancer agents on inhibiting HepG2 cell proliferation, adhesion, invasion, and migration, promoting cancer cell apoptosis. At lower concentrations, C20/C22 could effectively inhibit hepatic cancer cell proliferation while not affecting the cell viability of HUVEC and HEK293A, showing great potential for providing cancer-specific cytotoxicity effects without harming normal cells within a certain concentration range (Figure 2C).

Treatment with C20/C22 at the concentration (1 µM) that did not affect the growth of HepG2 cells but inhibited HepG2 cell migration (Figure 2F–I) and invasion (Figure 2E) indicated that the anti-migration and invasion effects were not due to the inhibition of cell growth. The critical step involved in cancer cell metastasis is the binding of cancer cells to vascular endothelial cells during circulation in the bloodstream [60]. Herein, the adhesion of HepG2 to HUVEC was significantly reduced by the treatment with C20/C22 (Figure 2D). This cancer cell–vascular endothelial cell interaction can be assisted by the interactions between selectins that are expressed on the vascular endothelium and the selectin ligand presented on cancer cells [61]. Therefore, reducing the expression of selectin ligand on cancer cells can effectively prevent the binding of cancer cells to endothelium [62]. Since C20 and C22 are glycosyltransferase inhibitors that can selectively inhibit the glycosyltransferases which are responsible for the synthesis of selectin ligands such as sialyl-Lewis structures [41], the ability of C20/C22 to modify HepG2 cell glycosylation was evaluated with lectin staining and immunofluorescence assays using lectins and carbohydrate antibodies. The binding of RCA I (recognized Gal residue), PHA-L (recognized Galβ4GlcNAcβ6), and MAL I (recognized Galβ4GlcNAc) to C20/C22-treated HepG2 was significantly reduced, which could be due to the inhibition of GalT activities responsible for transferring Gal to GlcNAc residues in the synthesis of the Galβ4GlcNAc structure (Figure 3). In addition, as the presence of Gal residues was reduced, the underlying GlcNAc could be exposed, thus enhancing the binding of WGA (recognized GlcNAc). These glycosylation alterations may require further glycomic analysis; in this study, we focused on the changes of outer chain epitopes. The results showed that treatment with C20/C22 led to the increased expressions of SLe^x^ and Le^y^ accompanied by the decreased expressions of Le^a^ and Le^b^ (Figure 4A–D). This could be due to the fact that ꞵ4GalT and ꞵ3GalT belong to the GalT family, and they recognize similar acceptor substrates and the same donor. In particular, ꞵ3GalT catalyzes the synthesis of Le^a^ and Le^b^, while ꞵ4GalT catalyzes the synthesis of Le^x^ and Le^y^ [63]. ꞵ4GalT and ꞵ3GalT can be competitors for the same donor and acceptor substrates. The results suggested that the inhibition of ꞵ4GalT by C20/C22 led to an increased production of the ꞵ3GalT-catalyzed products in HepG2 cell cultures. SLe^x^ is the ligand for P- and E-selectin [53,64], and their interactions mediate the spread and metastasis of cancer [47,54,55]. C20/C22-treated HepG2 cells exhibited reduced binding to the P- and E-selectin-coated plate (Figure 4E–H), suggesting the capability of C20/C22 to inhibit cancer metastasis. Furthermore, the removal of sialic acid by neuraminidase also disrupted the binding of the glyco-ligand to selectins and, thus, significantly decreased the adhesion ability (Figure 4I,J), indicating the sialic acid residue on the SLe^x^ epitope is critical for the selectin ligand interaction.

Cancer cells usually exhibit altered glycosylation, which is associated with defects in apoptosis [9]. Death receptors such as Fas and DR4/5 are glycoproteins which impact glycosylation during receptor expression, distribution, ligand–receptor interaction, and downstream signaling transduction [9]. Through binding to DR4/5, TRAIL triggers cell death through the extrinsic apoptotic pathway [65]. TRAIL is considered a highly promising anticancer drug, but the inherent tumor drug resistance has greatly limited the applicability of TRAIL-based therapy [66]. Our results showed that cultured HepG2 cells were resistant to TRAIL-induced apoptosis (Figure 5E,F). Treatment with C20/C22 could significantly increase the sensitivity of HepG2 to TRAIL-induced apoptosis. The percentage of apoptotic cells was elevated in comparison to the groups treated with only TRAIL or C20/C22. This could be attributed to the modification of cell glycosylation, which affected the expression and cell surface distribution of DR4 and DR5 (Figure 5A–D). After treatment with C20/C22, increased expressions of DR4/5 intracellularly and on cell surfaces were both observed via immunofluorescence assays (Figure 5C,D). In contrast, although Fas expression was affected by the treatment with C20/C22, the cell surface distribution of Fas remained unchanged (Figure 5G–I). As expected, the CH11-induced apoptosis was not affected by C20/C22 treatment (Figure 5J,K), since the increased expressions of Fas did not reach the cell surface where it could form a complex with the Fas ligand upon induction.

ER is a functional organelle responsible for protein folding and quality control. Protein glycosylation assists the protein folding process, and the accumulation of unproperly folded proteins can lead to ER stress, which in turn affects cell proliferation, apoptosis, and other biological processes [66]. Tunicamycin is a typical glycosyltransferase inhibitor which can block *N*-linked glycosylation, thus disrupting protein maturation and induces ER stress and apoptosis [67,68]. In this study the glycosyltransferase inhibitor C20/C22 also caused ER stress and an apoptosis-inducing effect in cultured HepG2 (Figure 6 and Figure 8). As indicated in the WB analysis (Figure 6A,B), the expressions of ER stress marker proteins GRP78, IRE1, ATF6, and PERK, as well as the phosphorylation of the downstream JNK, were all increased with the prolonged C20/C22 administration time. In accordance with other studies [69], C20/C22-induced ER stress also led to a cell-cycle arrest at the G2/M phase (Figure 6C,D).

C20/C22-treated HepG2 cells showed increased expression of the gene regulator CHOP (Figure 6A,B) which was reported to be highly produced under ER stress [70]. Consistent with other studies, our results showed that increased CHOP expression could upregulate the expression of proapoptotic proteins Bax and Bak (Figure 7A,B) and downregulate the expression of antiapoptotic protein Bcl-2. Further evidence has shown that the increased Bax/Bak expression leads to the formation of a complex that becomes embedded in the inner mitochondrial membrane, thus resulting in a decrease in the MMP and the release of cytochrome c and ROS [71]. Our results showed that the treatment of C20/C22 led to decreased MMP and an increased level of intracellular ROS, thus suggesting that C20/C22 treatment caused ER stress which eventually led to mitochondrial dysfunction (Figure 7C,D). The overproduction of intracellular ROS could further enhance the ER stress and cause cell-cycle arrest, as reported previously [72,73]. Mitochondrial dysfunction due to the treatment with C20/C22 led to an increased release of cytochrome c, which eventually activated the apoptosis signaling cascade (Figure 8) through an intrinsic pathway, indicating that C20/C22 exhibits apoptosis-inducing effects in cultured HepG2 cells.

The in vivo experiments with a xenograft mouse model further confirmed the antitumor activity of C20 and C22 (Figure 9 and Figure 10). Administration of C20/C22 inhibited tumor growth and angiogenesis, potentially preventing tumor metastasis in xenograft mouse model, while the same dosage of C20/C22 treatment did not cause weight loss or histopathological changes, such as deformity, swelling, or bleeding in major organs, showing their advantages to be developed as new anticancer drugs with low toxicity.

## 4. Materials and Methods

### 4.1. Materials

Rabbit polyclone anti-CHOP, cytochrome c (Cyto C), ATF6, PERK, GRP78, ARF4, Bax, Bak, caspase3, cleaved caspase 3, caspase 8, caspase 9, IRE1, DR4, DR5, P-JNK, Bcl-2, Fas, β-Actin, Gapdh, and Tubullin were obtained from Affinity Biosciences (Shanghai, China). Mouse monoclonal anti-Lewis a (Le^a^), Lewis b (Le^b^), and Lewis y (Le^y^) were purchased from Abcam (Cambridgeshire, UK) [74]. Mouse monoclonal anti-SLe^x^ was purchased from BD Biosciences (San Diego, CA, USA) [75]. FITC-conjugated anti-mouse/rabbit IgG was purchased from Affinity Biosciences (Shanghai, China). Goat anti-rabbit IgG-HRP was obtained from Life Science. E-selectin and P-selectin were purchased from R&D System (Minneapolis, MN, USA). Lectins were purchased from Vector Laboratories (Burlingame, CA, USA). Cells were cultured with cell growth medium (Gibco, Shanghai, China) containing fetal bovine serum (FBS) (ABW, Shanghai, China) and penicillin/streptomycin (Gibco, Shanghai, China).

### 4.2. Cell Cultures

Hepatocellular carcinoma cells (HepG2 and SMMC7721), were obtained from Suzhou Ruibo Biotechnology Co., Ltd (Suzhou, China). The cell culture conditions were 37 °C humidified with 5% CO_2_, and the cells were cultured in DMEM high-glucose medium containing 10% FBS, 1% penicillin G (60 mg/L), and streptomycin (100 mg/L). For experimental purposes, cells were seeded in culture flasks, culture plates, confocal small dishes (from Nest Biotechnology Co., Suzhou, China, Ltd.), and Transwell chambers (purchased from Corning Biotechnology Co., Ltd., Shanghai, China).

### 4.3. Cell Viability

Cells were seeded in a 96-well plate at a density of 8 × 10^3^ cells per well and cultured in complete growth medium for 24 h. After 24 h, the cells were washed with precooled PBS, and the cells were treated with C20 or C22 at different concentrations for 24 h. Dimethyl sulfoxide (DMSO) was added to the control group at the same concentration as was added to the drug treatment group. The cell viability test was carried out according to the CCK8 kit (Yeasen, Shanghai, China) instructions.

### 4.4. Adhesion to HUVEC Cells

After the HepG2 cells were treated with 2 μM C20 or C22 for 24 h, the viable cells were counted with Trypan blue (Beyotime Biotechnology, Shanghai, China), and then these HepG2 cells were stained with Calcein AM (Yeasen, Shanghai, China). A total of 2 × 10^4^ stained HepG2 cells from the control and C20/C22-treated cells were added per well to a 96-well plate covered with a single layer of HUVEC cells, before incubating at 37 °C for 2 h. These samples were then washed with precooled PBS three times to remove the unadhered cells, and a fluorescence microplate reader was used to detect the fluorescence intensity (with an excitation wavelength of 488 nm and an emission wavelength of 525 nm).

### 4.5. Cell Invasion Assays

The invasion ability of HepG2 cells was evaluated by transwell assays. Briefly, the logarithmic growth cells were seeded in six-well plates, and the cells were treated with different concentrations of C20 or C22 for 24 h; then, the live cells were digested, resuspended, and counted. The cells were loaded into the inserts that were coated with Matrigel (BD Biosciences, San Diego, CA, USA) (dilution ratio 1:30) at a density of 4 × 10^5^ cells/well. Next, 600 μL of medium containing 10% FBS was added to the bottom space of the chamber. After 24 h of incubation, the cells on the upper layer of the inserts were removed, and the cells on the lower layer were fixed with 95% ethanol and stained with crystal violet. The cell invasion behaviors were recorded by an inverted microbial microscope.

### 4.6. Wound Healing

Cells were incubated in a six-well plate. When the convergence was approximately 85%, a 20 μL pipette tip was used to draw a wound on the bottom of the cell culture plate. The plate was washed with PBS to remove the fallen cells, and then different concentrations of C20 or C22 solution were added to treat the cells for 24 h. The same concentration of DMSO was added to the cell culture and used as the control group. Subsequently, the cell medium was changed to the complete medium containing 4% serum. Photographs of the wound were recorded at 0, 24, and 48 h. ImageJ calculated the scratch width and migration distance, while the statistical differences between groups were determined via one-way ANOVA followed by Tukey’s test calculations.

### 4.7. Immunofluorescence

Cells were seeded in a 24-well plate containing glass cell slides, and C20 or C22 were added to treat the cells for 24 h when the reconciliation degree reached over 85%. After the treatment, cells were washed 2–3 times with precooled PBS, and then fixed with 4% paraformaldehyde for 10 min at room temperature (RT); they were subsequently washed three times with PBST. To stain the intracellular proteins, the cell surface was perforated via incubation with 0.25% Triton × 100 for 10 min at RT. To stain the cell surface proteins, 2.5% BSA was directly added, and the sample was sealed at RT for 1 h. After blocking, FITC-labeled primary antibodies or unconjugated primary antibodies (dilution ratio: 1:200) were added and incubated overnight at 4 °C [76]. After the primary antibodies were incubated, they were washed three times with PBST for a total time of 15 min; then, FITC-labeled goat anti-rabbit or anti-mouse immunoglobulin (dilution ratio: 1:300) was added and incubated for 1 h at RT. After washing three times with PBST, DAPI was added to stain the nucleus for 1 min. The expression of the target protein was observed with the laser confocal microscope.

### 4.8. Selectin Recognition Analysis

First, ELISA plates were coated with E- or P-selectin overnight at 4 °C, and 1% BSA blocking solution was added and kept at 4 °C overnight. After blocking, 2 × 10^4^ live cells pretreated with C20 or C22 were added to each well; the same concentration of DMSO solution was added to the cell culture and employed as a control to compare with the C20 or C22 treatment. Incubation was performed at 37 °C for 2 h, followed by fixation with 10% formaldehyde and air-drying. The adherent cells were stained with 0.05% crystal violet at RT for 10 min, and they were then washed in water several times until the dye stopped falling off from the cells. The cell adhesion was recorded under a microscope; decolorizing solution (50% ethanol, 0.1% acetic acid) was added to the wells, and the absorbance was read at 590 nm using a microplate reader.

### 4.9. Lectin Staining Assays

Lectins were used to analyze cell surface glycosylation. The lectin staining was performed according to a previously described method [77,78]. Cells were firstly seeded into a 96-well plate, 2 μM C20 or C22 was added to treat the cells for 48 h, and the supernatant was discarded; then, different biotin-labeled lectins (1 µg/mL) were added to the specific wells prior to the addition of alkaline phosphatase-conjugated AVIDIN (Sigma-Aldrich, Shanghai, China) and nitrobenzene phosphate reaction substrate (Sigma-Aldrich, Shanghai, China). The absorbance values were recorded using a microplate reader at 405 nm. The absorbance values of Con A staining in the treatment group and the control group were used for normalization. The *t*-test method was used to calculate the difference between groups. A value of *p* < 0.05 was considered statistically significant. Each assay experiment was repeated at least six times.

### 4.10. Western Blot Analysis

Cultured cells were lysed on ice for 10 min in RIPA buffer (50 mM Tris-HCl (pH 7.4), 150 mM NaCl, 1% NP-40, 0.1% SDS) with phosphatase inhibitors (250 mM sodium fluoride, 50 mM sodium pyrophosphate, 50 mM β-glycerophosphate, and 50 mM sodium orthovanadate in H_2_O) and protease inhibitors (AEBSF. HCl, Aprotinin, Bestatin, E-64, Leupeptin, Pepstatin A). A small amount of protein sample was used for BCA protein quantification. SDS protein loading buffer was added to the remaining protein samples, and these samples were then boiled at 95 °C for 10 min. The protein was separated via polyacrylamide gel electrophoresis and transferred to a PVDF membrane. It was then blocked with a TBST solution containing 5% skimmed milk powder at room temperature for 2 h. This membrane was incubated overnight at 4 °C with primary antibodies. The titer of the primary antibodies used in this study may vary from 1:500 to 1:2000. We used 1:1000 dilution for all primary antibodies as suggested by the manufacturer’s instructions. Subsequently, the membrane was washed three times with TBST (10 min each time) and incubated with horseradish peroxidase-labeled goat anti-rabbit or anti-mouse IgG (dilution ratio: 1:10000) for 2 h at RT. After washing three times (10 min each time). The Western blot was visualized by enhanced chemiluminescence. The protein bands were quantified with ImageJ software.

### 4.11. Mitochondrial Membrane Potential Assay

The method for mitochondrial membrane potential detection was carried out according to the kit instructions. Briefly, after 24 h of drug treatment (*n* = 3), freshly prepared JC-1 staining solution (Beyotime Biotechnology, Shanghai, China) was added to stain the cells at 37 °C for 2 h, and 1 μM CCCP (Beyotime Biotechnology, Shanghai, China) was used as a positive control. After staining, the washing buffer provided in the kit was used to remove unbound JC-1 staining solution. Distributions of JC-1 monomer (green fluorescence) and JC-1 polymer (red fluorescence) were detected by fluorescence microscopy to determine the loss of mitochondrial membrane potential in the C20 or C22 administration groups compared with the control group.

### 4.12. Active Oxygen Species (ROS) Detection

HepG2 cells were seeded in six-well plates. When the cell density reached 85% or more, different concentrations of C20 or C22 (2, 3 μM) were added to treat the cells for 24 h. The cells were digested, washed with precooled PBS, and stained with 10 μM DCFH-DA (Sigma-Aldrich, Shanghai, China) at 37 °C for 30 min, and then free DCFH-DA was removed by washing with PBS. The level of ROS was detected using fluorescence microscopy.

### 4.13. Cell-Cycle Analysis

In order to evaluate the distribution of HepG2 cancer cells at different stages of the cell cycle, approximately 5 × 10^5^ cells/well were incubated in a six-well plate overnight at 37 °C to allow the cells to adhere. The cells were then treated with different concentrations of C20 or C22 for 24 h. Pyridine iodide (PI) staining was performed according to the instructions provided with the kit. Flow cytometry was used to detect the distribution of HepG2 cells in different cell-cycle stages; analysis was performed using Modfit LT5 software, and one-way ANOVA followed by Tukey’s test was used for statistical analysis.

### 4.14. Cell Apoptosis Tests

An Annexin V–FITC/PI double staining method was used to detect cell apoptosis. A total of 5 × 10^5^ cells were seeded in a six-well plate. The cells were treated with C20 or C22 (2, 3 μM) for 24 h, and all cells in the well were collected. These cells were washed three times with precooled PBS and centrifuged at 300× *g* for 5 min at 4 °C each time. Finally, Annexin V-FITC/PI dye (Yeasen, Shanghai, China) was added according to the kit instructions, the cells were stained for 15 min at 37 °C in the dark, and the apoptotic cells were detected via flow cytometry. For TRAIL and Fas L inducing apoptosis, 30 ng of TRAIL (Cedarlane Ontario, Canada) or 100 ng of anti-Fas antibody (CH11) (Milipore, Shanghai, China) was added to each well after cells were treated with 2 μM C20 or C22 for 24 h, and the remaining steps were similar to those described above.

### 4.15. In Vivo Toxicity and the Antitumor Effect of C20/C22

The 6–8 week old BALB/c nude mice were purchased from Beijing Weitong Lihua Biotechnology Co., Ltd (Beijing, China). Animal experiments performed in this study are widely used and were reviewed by a Research Ethics Committee of Jilin University (Reference number: S2021086). An acclimation period of 7 days (excluding the arrival date) prior to the experiments allowed the animals to stabilize in the new environment. The mice were group-housed (five animals per cage) in normalized conditions (room temperature (23 ± 2) °C, half day and night, 8:00 a.m.–8:00 p.m. in light); all animals could eat and drink freely.

To evaluate the toxicity of C20/C22, healthy BALB/c nude mice (total of 21 in seven groups, *n* = 3) were administrated with C20/C22 according to the dosage shown in Figure 9A. C20/C22 was administered once every 2 days by subcutaneous (sc) injection during the first week, before changing to one injection every 3 days after the first week of administration. PBS solution containing 0.5% ethanol (blank) was used as the control group. Body weights of mice were recorded on each injection day. Mice were euthanized on day 24.

To establish the xenograft mouse model (total of 12 in three groups, *n* = 4), BALB/c nude mice were subcutaneously inoculated with 1 × 10^7^ HepG2 cells in a 100 μL cell suspension containing 50% matrix glue (BD Biosciences, San Diego, CA, USA) on the right back. After 15 days when tumors were palpable and the tumor volume reached about 100 mm^3^, 5 μmol/kg of C20/C22 was administered by sc injection, and the model group was injected with PBS solution containing 0.5% ethanol. The mice were administered with C20/C22 once every 2 days during the first 8 days of treatment and then switched to one injection every 3 days. Tumor volume (V = 0.5 × ab^2^), where a is the long axis of the tumor, and b is the short axis of the tumor, measured with a vernier caliper, Germany Manette, IP54) and body weight were recorded on all administration days. The mice were euthanized once the tumor volume of the model group reached 1000 mm^3^, and the size and weight of tumors were recorded. The major organs from mice were stained by hematoxylin and eosin (H&E) for histopathology analysis, and the blinded histopathology evaluation was performed by a pathologist.

Paraffin sections of the major organs were subjected to the staining with antibodies of GRP78 (1:100), Cyto C (1:50) and cleaved caspase 3 (1:200) to further assess the toxicity of C20/C22. Paraffin sections of the tumor tissue were stained with primary antibodies including GRP78 (1:100), Cyto C (1:50), Ki67 (1:200), cleaved caspase 3 (CC3, 1:200), DR4 (1:100), DR5 (1:100), and SLe^x^ (1:100) for immunohistochemical analysis [79]. Sections were scored by counting the number of cancer cells expressing the marker protein CD31. Other sections were assessed by H-Score [80]. Briefly, the positive intensity of staining was multiplied by the positive ratio to obtain a semiquantitative H-Score. Here, 0–25%, 26–50%, 51–75%, and 76–100% were scored as 1–4, respectively, according to the positive ratio. None, weak, medium, and strong were scored as 1–4 according to the staining intensity.

## 5. Conclusions

In summary, we studied the anticancer effects of synthesized glycosyltransferase inhibitor C20/C22 using the HepG2 cell line and a xenograft mouse model. C20/C22 could effectively inhibit the growth of HepG2 while showing less cytotoxicity to normal cells (HUVEC and HEK293A). C20/C22 treatment led to the alteration of cell glycosylation, and it reduced selectin ligand (i.e., SLe^x^)-mediated cancer cell–endothelium adhesion and metastasis. In addition, C20/C22 affected the intracellular expression and cell surface distributions of DR4 and DR5, as well as increased the sensitivity of HepG2 to TRAIL-induced apoptosis C20/C22, thus showing potential as an anticancer adjuvant to resume the susceptibility of TRAIL-resistant cancer cells to TRAIL-based anticancer drugs. Furthermore, C20/C22 treatment induced ER stress in HepG2 cells, led to cell-cycle arrest and mitochondrial dysfunction, and finally stimulated the caspase cascade reaction, thus leading to cell apoptosis. In vivo investigation of the antitumor effects of C20/C22 with xenograft mouse model showed the increased expression of ER stress, mitochondrial dysfunction, and apoptosis markers, as well as reductions in angiogenesis and tumor cell growth up on C20/C22 treatment. We observed reduced tumor volume and tumor weight after the treatment without affecting the major organs, further confirming the strong potential of C20/C22 as a candidate for development as a new generation of anticancer agents for the treatment of malignant tumors with low toxicity. We also demonstrated that C20/C22 can be used in combination with TRAIL-based cancer therapy to overcome the multidrug resistance in some cancers.

## Figures and Tables

**Figure 1 pharmaceuticals-15-00716-f001:**
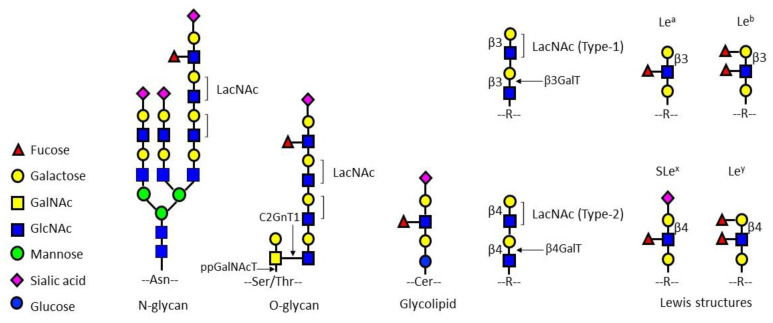
Structures of LacNAc and Lewis epitopes present in *N*/*O*-glycans and glycolipids.

**Figure 2 pharmaceuticals-15-00716-f002:**
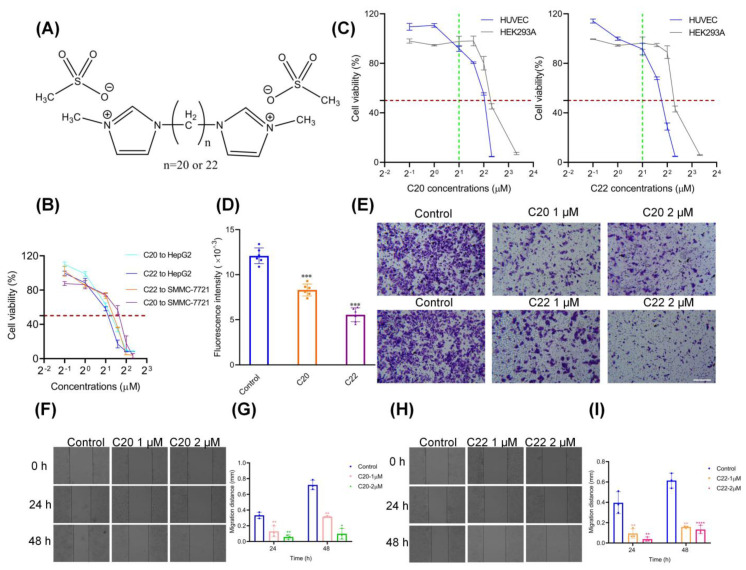
C20 and C22 inhibited HepG2 growth and adhesion to HUVECs, as well as prevented HepG2 invasion and metastasis. (**A**) Structure of C20/C22. (**B**) The growth-inhibitory effect on HepG2 and SMMC-7721 cells in response to treatment with C20/C22 at various concentrations for 24 h (*n* = 3). (**C**) The effect of C20/C22 treatment on the viability of HUVEC and HEK293A cells (*n* = 3). (**D**) C20/C22 (2 μM) affected the adhesion of HepG2 cells to HUVECs (*n* = 6). (**E**) C20/C22 hindered the trans-endothelium invasion of HepG2 cells (*n* = 3). Scale bars, 50 µm. (**F**,**G**) HepG2 cells were treated with 1 and 2 μM C20 for 24 and 48 h, and the cell migration distance results were calculated after this treatment (*n* = 3). (**H**,**I**) HepG2 cells were treated with 1 and 2 μM C22 for 24 and 48 h, and the cell migration distance results were calculated following this treatment (*n* = 3). The values shown are the mean ± SD of three independent experiments. The *p*-value was analyzed by one-way ANOVA followed by Tukey’s test using GraphPad Prism version 8.00. * *p* < 0.05, ** *p* < 0.01, *** *p* < 0.001 and **** *p* < 0.0001 vs. control group.

**Figure 3 pharmaceuticals-15-00716-f003:**
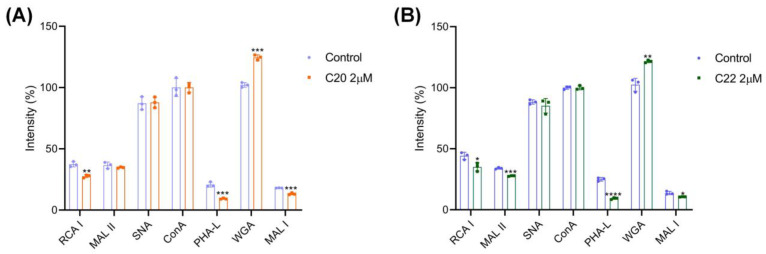
Lectin staining analysis of cell surface glycosylation. (**A**,**B**) Lectin staining of HepG2 cells after treatment with C20/C22 at a concentration of 2 μM for 48 h (*n* = 3). The displayed data correspond to the mean ± SD of three separate experiments. The *p*-value was analyzed by Student *t*-test using GraphPad Prism version 8.00. * *p* < 0.05, ** *p* < 0.01, *** *p* < 0.001 and **** *p* < 0.0001 *vs.* control group.

**Figure 4 pharmaceuticals-15-00716-f004:**
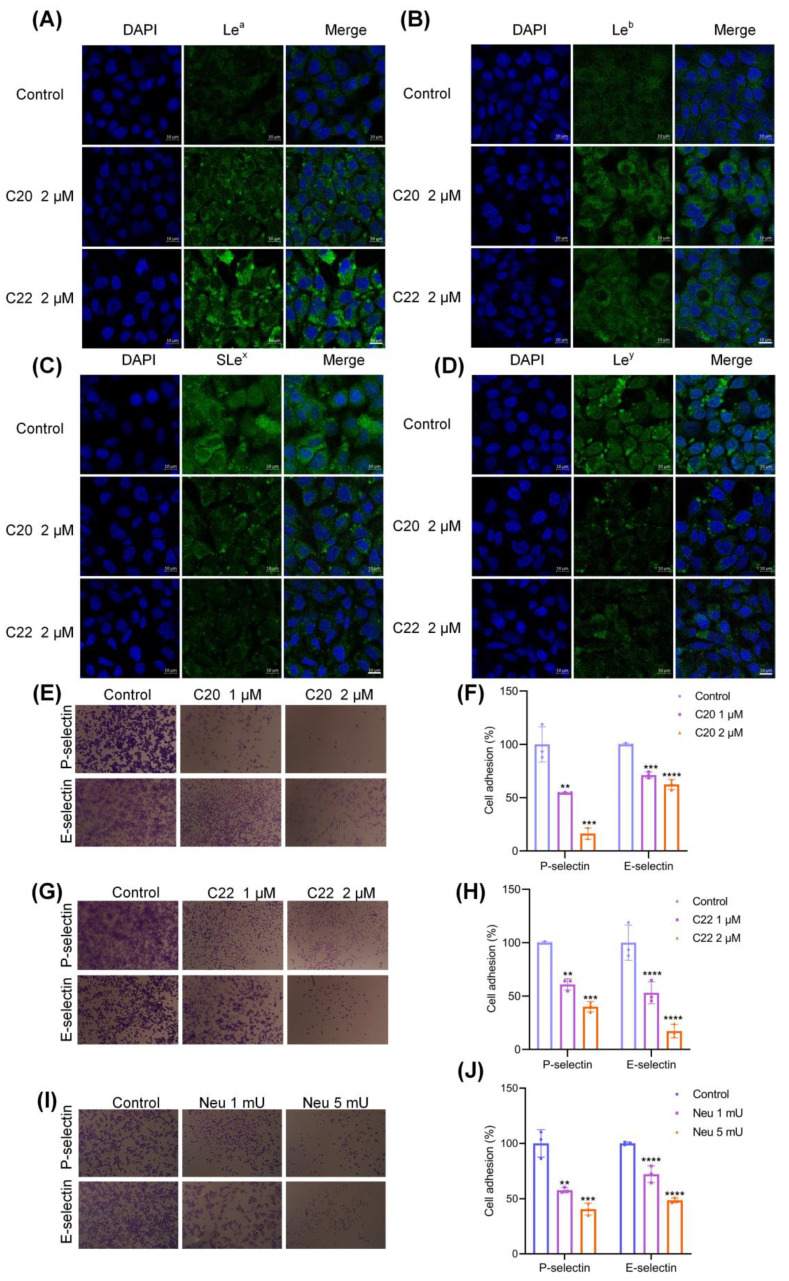
C20 and C22 modified cell surface glycosylation and inhibited the interactions between HepG2 and E/P-selectin. HepG2 cells were treated with 2 μM C20/C22 for 24 h, and the expression of Lewis oligosaccharides on the surface of HepG2 cells was detected by immunofluorescence assays: (**A**) Le^a^, (**B**) Le^b^, (**C**) SLe^x^, and (**D**) Le^y^. Scale bars, 10 µm. Different concentrations of C20, C22, and neuraminidase were used to treat cells for 24 h, and then 2 × 10^4^ live cells/well were taken and added to the P-selectin- and E-selectin-coated ELISA plates (*n* = 3). (**E**,**F**) The effect of treatment with 1 and 2 µM C20 on cell adhesion to P- and E-selectin. (**G**,**H**) The effect of treatment with 1 and 2 µM C22 on cell adhesion to P- and E-selectin. (**I**,**J**) The effect of 1 and 5 mU neuraminidase on cell adhesion to P- and E-selectin. Scale bars, 50 µm. The displayed data correspond to the mean ± SD of three separate experiments. The *p*-value was analyzed by one-way ANOVA followed by Tukey’s test using GraphPad Prism version 8.00. ** *p* < 0.01, *** *p* < 0.001, and **** *p* < 0.0001 *vs.* control group.

**Figure 5 pharmaceuticals-15-00716-f005:**
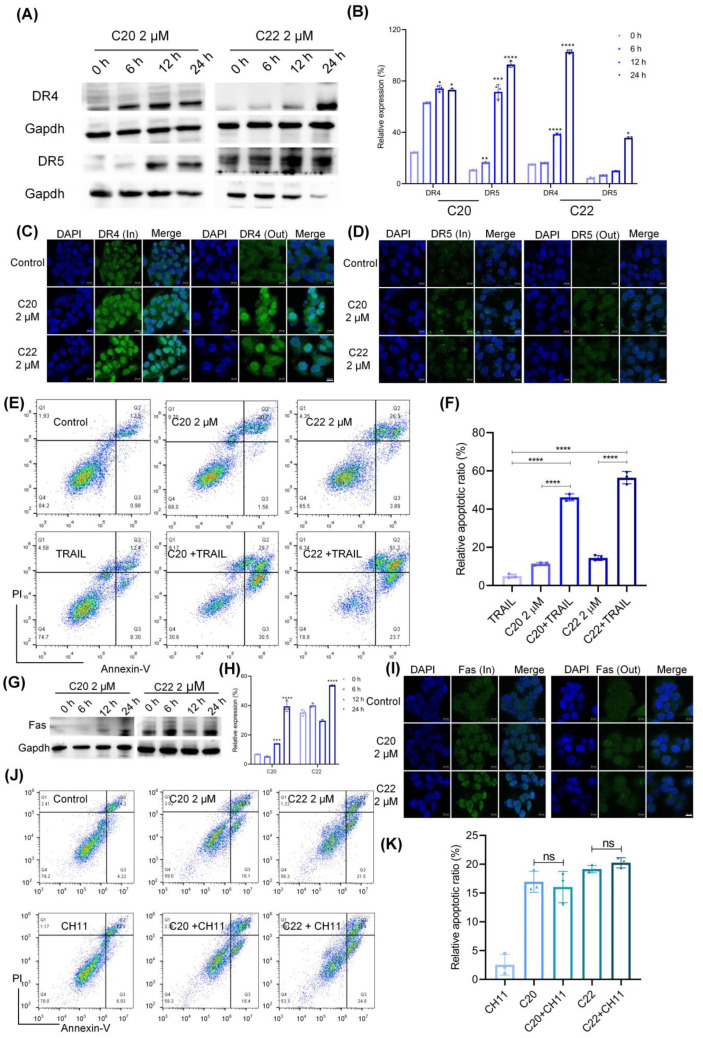
C20/C22 increased the sensitivity of HepG2 cells to TRAIL-induced apoptosis. (**A**,**B**) WB analysis of the expression of DR4 and 5 in HepG2 cells that were treated with 2 μM C20/C22 for 0, 6, 12, and 24 h. ImageJ software was used to calculate the gray value (*n* = 3). (**C**,**D**) The immunofluorescence assays used to determine the intracellular and cell surface distribution of DR4/5. (**E**,**F**) Annexin V/PI double staining was used to detect the apoptotic cells after the indicated treatments (*n* = 3). (**G**,**H**) WB analysis of the expression of Fas in HepG2 cells treated with C20/C22 for 0, 6, 12, and 24 h. (**I**) The immunofluorescence assays used to determine the intracellular and cell surface distribution of Fas. (**J**,**K**) Flow cytometry analysis of the apoptotic cells after the indicated treatments (*n* = 3). Scale bars, 10 µm. Data shown correspond to the mean ± SD of three independent experiments. The *p*-value for all datasets was analyzed by one-way ANOVA followed by Tukey’s test using GraphPad Prism version 8.00, except the data in (**B**) whose *p*-value was analyzed by nonparametric Dunnett’s test using GraphPad Prism version 8.00. * *p* < 0.05, ** *p* < 0.01, *** *p* < 0.001, **** *p* < 0.0001, and ns nonsignificant *vs.* 0 h/control group.

**Figure 6 pharmaceuticals-15-00716-f006:**
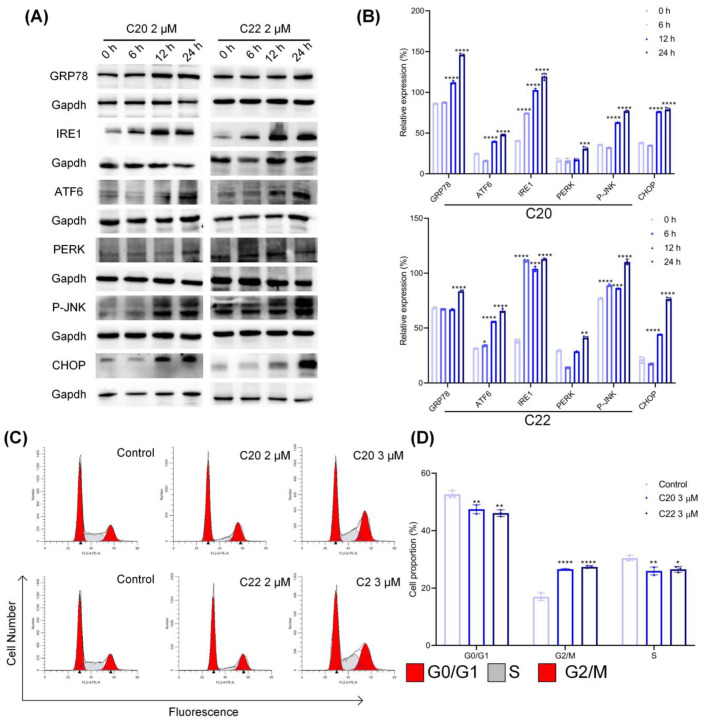
Treatment with C20/C22 induced ER stress and cell-cycle arrest of HepG2. (**A**,**B**) Expressions of ER stress-associated proteins in HepG2 cells after treatment with 2 μM C20/C22 for 0, 6, 12, and 24 h. All proteins were normalized with the expression of Gapdh (*n* = 3). (**C**,**D**) Cells were stained with PI after 24 h of C20/C22 incubation, and flow cytometry was used to determine the proportion of cells in each stage of the cell cycle (*n* = 3). The reported data correspond to the mean ± SD of three independent experiments. The *p*-value was analyzed by one-way ANOVA followed by Tukey’s test using GraphPad Prism version 8.00. * *p* < 0.05, ** *p* < 0.01, *** *p* < 0.001, and **** *p* < 0.0001 *vs.* 0 h/control group.

**Figure 7 pharmaceuticals-15-00716-f007:**
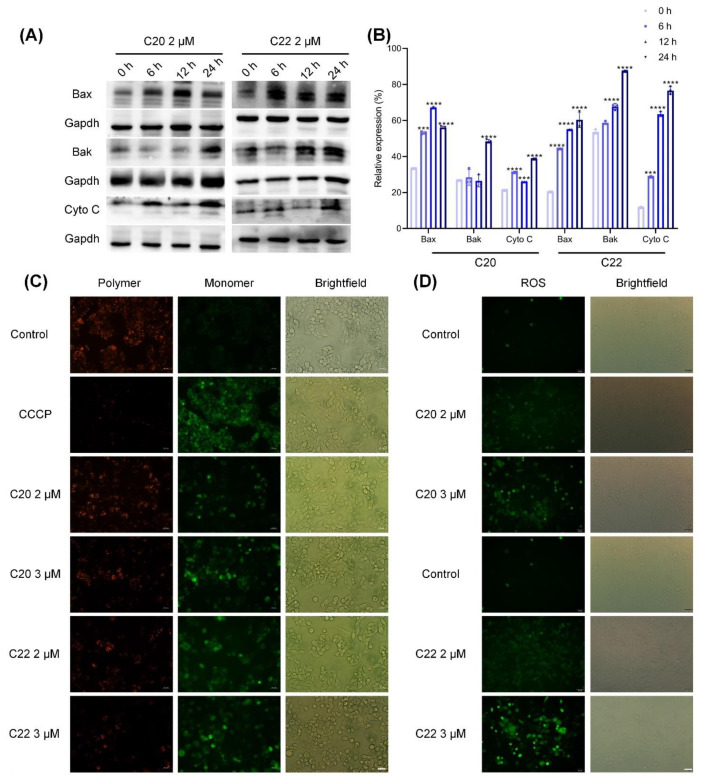
Treatment with C20/C22 induced ER stress and mitochondrial dysfunction increased the expression of proapoptotic proteins. HepG2 cells were treated with 2 and 3 μM of C20/C22 for 24 h. (**A**,**B**) WB analysis of the protein expression in HepG2 cells treated with 2 μM of C20/C22 for 0, 6, 12, and 24 h. The presented values correspond to the mean ± SD for three independent experiments. The *p*-value was analyzed by one-way ANOVA followed by Tukey’s test using GraphPad Prism version 8.00. *** *p* < 0.001, and **** *p* < 0.0001 *vs.* 0 h/control group. (**C**) JC-1 staining was performed to measure the MMP (*n* = 3), CCCP, and carbonyl cyanide 3-chlorophenylhydrazone. (**D**) C20/C22-treated cells were stained with DCFH-DA, and the relative levels of intracellular ROS were determined via fluorescence microscopy (*n* = 3). Scale bars, 50 µm.

**Figure 8 pharmaceuticals-15-00716-f008:**
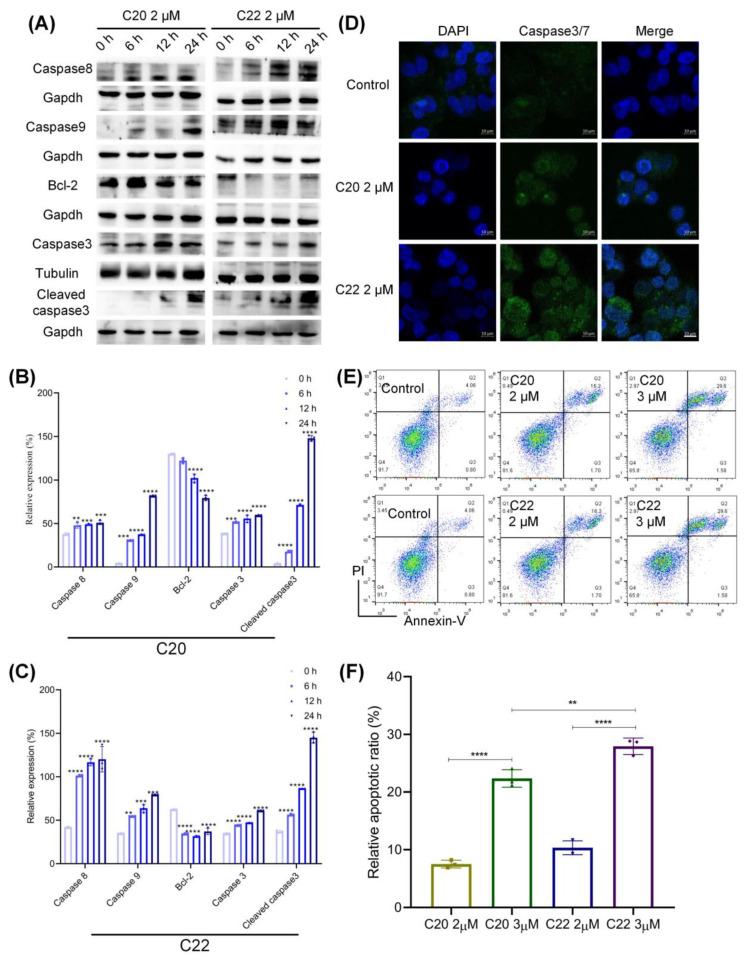
Treatment with C20/C22 led to increased proapoptotic protein expression and a caspase cascade response. (**A**–**C**) WB analysis of the antiapoptotic and caspase proteins in HepG2 cells treated with C20/C22 at 2 μM for 0, 6, 12, and 24 h. The reported values correspond to the mean ± SD for three independent experiments. (**D**) Immunofluorescence assays employed to determine the expression of caspase3/7 in cells with indicated treatments. Scale bars, 10 µm. (**E**,**F**) Apoptotic cells detected by flow cytometry after indicated treatments (*n* = 3). The *p*-value was analyzed by one-way ANOVA followed by Tukey’s test using GraphPad Prism version 8.00. ** *p* < 0.01, *** *p* < 0.001, and **** *p* < 0.0001 *vs.* 0 h/control group.

**Figure 9 pharmaceuticals-15-00716-f009:**
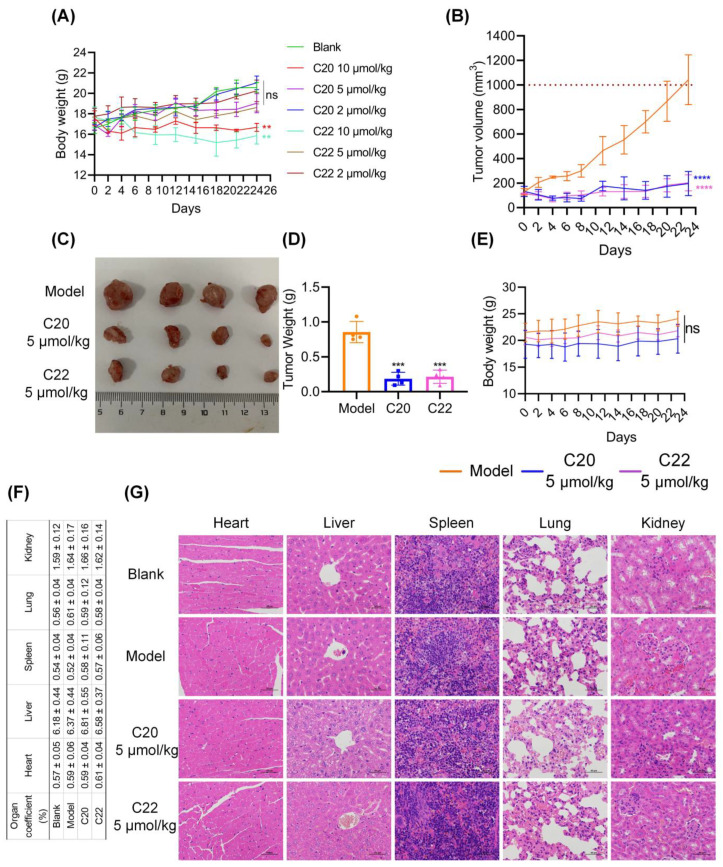
The toxicity and antitumor effects of C20/C22 in vivo. (**A**) In vivo toxicity study of C20/C22, showing the effects of C20/C22 on the body weight (*n* = 3). (**B**) Tumor volume of tumor-bearing mice until 24 days post administration (*n* = 4). (**C**) Tumor images at 23 days post administration. (**D**) Tumor weight at 23 days post administration. (**E**) Body weight of tumor-bearing mice until 23 days post administration. (**F**) Organ coefficient statistics. (**G**) Histopathology of major organs from xenograft mouse model post administration in comparison to the healthy mice (H&E stain × 400). Scale bars, 50 µm. The *p*-value was analyzed by one-way ANOVA followed by Tukey’s test using GraphPad Prism version 8.00. ** *p* < 0.05, *** *p* < 0.001, **** *p* < 0.0001 and ns nonsignificant *vs.* blank/model group.

**Figure 10 pharmaceuticals-15-00716-f010:**
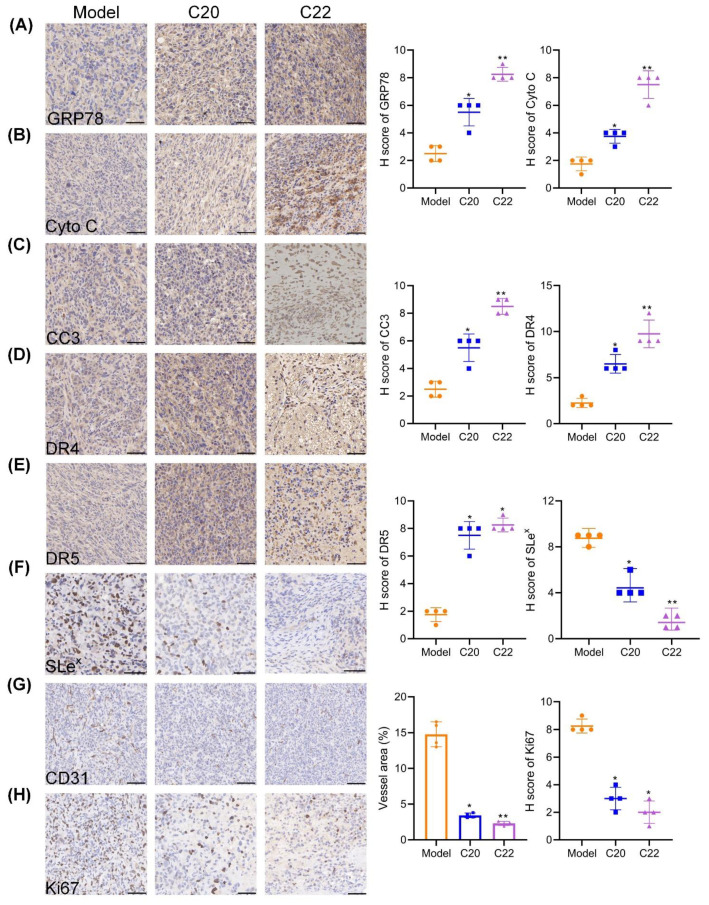
Immunohistochemical analysis of tumor sections from model group and the groups administrated with C20/C22. (**A**) GRP78 staining tumor sections (*n* = 4). (**B**) Cyto C staining tumor sections (*n* = 4). (**C**) CC3 staining tumor sections (*n* = 4). (**D**,**E**) Death receptors DR4 and DR5 staining tumor sections (*n* = 4). (**F**) SLe^x^ staining tumor sections (*n* = 4). (**G**) CD31 staining tumor sections (*n* = 4). (**H**) Ki67 staining tumor sections (*n* = 4). Scale bars, 50 μm. Data are presented as the mean ± SD. The *p*-value was analyzed by nonparametric Dunnett’s test using GraphPad Prism version 8.00. * *p* < 0.05, ** *p* < 0.01 treatment groups *vs.* model group.

## Data Availability

Data is contained within the article and Appendix A.

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
