# Peer review of "Bisimidazolium Salt Glycosyltransferase Inhibitors Suppress Hepatocellular Carcinoma Progression In Vitro and In Vivo"

_pharmaceuticals, 2022, doi:10.3390/ph15060716_

Round 1

Reviewer 1 Report

Overall, this is a detailed and extensive study of the biological and biochemical impacts of the bisimidazole inhibitors C20 and C22. The data presented provide evidence of not only the ability of these compounds to reduce tumor size, while also reducing cancer cell migration. The paper has a lot of interesting and compelling data and warrants publication. My minor critiques are that there are a few instances of english language issues (including line 12 and line 43, but also elsewhere), so please carefully review the text. Also, it would be helpful to include a figure that corresponds to the various carbohydrate compounds described in the introduction 

Reviewer 2 Report

In this study, the authors investigated the efficacy of bisimidazolium salts as glycosyltransferase inhibitors to decrease hepatocellular carcinoma (HCC) progression in vitro and in vivo. This study seems interesting, however there are a number of issues with the manuscript that the author should consider :

- It is unfortunate that this study was performed only on a single hepatic cancer cell line (HepG2). Why not have used other HCC cell lines (at least a 2nd one) to confirm the results obtained on HepG2 cells ?

- There is no information about the glycosyltransferase inhibitors C20/C22 used in this study. This point should be clarified and data added in the introduction and/or discussion sections.

- Lectin staining results shown in figure 2, are not very convincing. Experiments using flow cytometry, a more robust technique to semi-quantify differences in lectin stainings, should be necessary to confirm the data.

- To confirm the results of Lewis antigen expression obtained by immunofluorescence, it would be important to determine also the expression of sialyl-lewis a and lewis x antigens at the surface of HepG2 cells upon C20/C22 treatment. Results obtained by flow cytometry coud be better to semi-quantify the expression of Lewis antigens.

-  It is necessary to give the references of anti-Lewis antibodies used in this study, in material and methods section. Suppliers can provide many antibodies with varying degrees of robustness.

- The meaning of « CCCP » in figure 6 should be added in the legend of figure. There is also a discrepancy between the legend of figure 6 and the order of the panels A, B, C and D.

- The figures 4B, E, F, H, J K are too small to be correctly analysed, specifically the percentage of each sub-population of cells in 4E and J are illegibles. It is necessary to enlarge them to increase their clarities.

- The expression of GRP78, Cyto C and cleaved caspase 3 should also be determined in major organs to consolidate the H&E staining results shown in figure 8.

- There is a discrepancy between the texte page 13 lines 338-345 and the results presented in figure 3 regarding the expression of Lewis antigens. Moreover the β3GalT seem to be rather activated. These points should be checked and clarified.

- To confirm the results showing a decrease in the binding of C20/C22-treated HepG2 cells to selectins in vitro, it would be important to determine the anti-metastatic effect of C20/C22 in vivo after orthotopic inoculation of HepG2 cells in nude mice.

- Some typographical errors must be corrected : e.g. : « Polymeride » in figure 6 ; « contorl » in figure 4I ; …

Reviewer 3 Report

The manuscript pharmaceuticals-1700912 entitled “Bisimidazolium Salt Glycosyltransferase Inhibitors Suppress Hepatocellular Carcinoma Progression in Vitro and in Vivo” submitted to MDPI section pharmaceuticals. It is an article that demonstrates with sufficient experiments the effect of C20 and C22 in suppressing the progression of hepatocellular carcinoma. Full membranes of the westerns blots are welcome to be attached. However, the following items need to be modified.

  • Check spelling. For example "concemtrations".
  • Statistical analysis. Were all analyses really performed with one way anova? In order to perform the anova test, normality and homogeneity of variances must first be checked. Have you performed this check? With an n=3 the data are more likely to be nonparametric. Please indicate that these tests have been passed.
  • The figures are too small, there are graphs that are difficult to read.
  • In a Q1 journal, the graphical representation should be more rigorous than simple bar graphs, since these only represent means and sd. Change to box plots for example or represent all points on the bar chart to be able to check the real dispersion, maxima and minima.
  • Check all the images because in some figures the scale bar is missing. Debe de haber al menos una scale bar en una imagen de cada figura. There must be at least one scale bar in one image of each figure.

Round 2

Reviewer 2 Report

-

Author Response

Thanks again to the reviewer for precious time in reviewing our paper.